# Bonds, Battles and Social Capital: Power and the Mediation of Water Insecurity in Peri-Urban Gurgaon, India

**Vishal Narain [1], Sumit Vij [2],\* and Aman Dewan [3]**

1   Management Development Institute, Gurgaon 122007, India
2   Public Administration and Policy Group, Wageningen University and Research, 6706 KN Wageningen, The Netherlands
3   Save the Children, 110065 New Delhi, India
\*   Correspondence: sumit.vij@wur.nl; Tel.: +31-682-56-1151

**Abstract:** This article describes the role of social capital and power as a significant underlying factor influencing water security in peri-urban Gurgaon. The article shows how differential access to social capital shapes differential access to water. In peri-urban contexts, communities that lack access to water mobilise their social capital to enhance their water security. We use the concepts of power and social capital to explain how the actors interact in peri-urban Gurgaon, paying attention to which social groups are powerful and how the powerless use social capital to adapt to changing resource access and usage. We conclude by drawing theoretical- and policy-relevant insights from the field.

**Keywords:** social capital; power; peri-urban; water security; Gurgaon; India

---

## 1. Introduction

Peri-urban areas are spaces in transition, combining features of both urban and rural environments. These spaces are institutionally and ecologically dynamic [1], characterized by socio-economic heterogeneity and growing competition over natural resources [2–5]. Peri-urban areas face the repercussions of urban expansion, providing the land, water and other resources for urban agglomerations and receiving urban waste [6]. Such changes in peri-urban spaces give rise to conflicts while also engendering new forms of cooperation [2]. Significant attention has been paid to the dimensions of peri-urban water insecurity, focusing on the myriad ways in which urbanization erodes the access of peri-urban communities to water [3,7]. Water security can be defined as the availability of acceptable quantity and quality of water for health, livelihoods, ecosystems and production [8]. With this definition, peri-urban scholars have sufficiently demonstrated how the growth of urban agglomerations has been aggravating water insecurity for household and agricultural uses in peri-urban environments [3,9,10].

However, the peri-urban literature does not adequately respond to the question of how peri-urban communities adapt to changing water access and control [2]. In fact, peri-urban research has paid little attention to the internal social and institutional dynamics shaping access to water in peri-urban spaces. The literature provides even fewer insights on the role of social capital and power relations among peri-urban communities in shaping access to water and in adapting to water insecurity [11]. At the same time, the peri-urban literature remains silent on the role of power relations among peri-urban communities in shaping access to water. Through this article, we seek to bridge these two gaps.

Social capital is relevant in peri-urban spaces for a number of reasons [12,13]. On the one hand, migration and occupational diversification consequent to land acquisition can erode the bases of

social interaction and weaken social capital. In the face of declining access to water and other natural resources, there is a need for scientific investigation of how peri-urban communities adapt to the changes underway, and in particular, on the role of social capital in the mediation of water insecurity. This article shows the different ways in which social capital is mobilized to improve access to water in a peri-urban context and how power is exercised differently by various social groups. It shows how differential social capital shapes access to water and how one particular community is able to overcome its poor access to water by mobilising its social capital. It touches upon how social capital is created by informal associations between people in the presence of more formal organizations and groups. By doing so, we seek to make the concepts of social capital and power central to discourses on peri-urban water security.

The remainder of the article is organized as follows: Section 2 elaborates the concepts of social capital and power and how the two concepts are operationalized in this article. Section 3 is the methodology section, elaborating the geographical context of the study and describing the methodology used to conduct the research. It is followed by Section 4, discussing the current water supply situation of the village, especially focusing on the role of village politics in influencing water supply, followed by a description of adaptive responses of the social groups which are not satisfied with the water supply situation. It also describes the historical loss of access to resources by the lower castes, specifically focusing on the ways through which the upper caste communities dominated and influenced the access to water of the dalits (communities belonging to the lower castes). We reflect upon the role of social capital in coping with water security, emphasizing on drinking water as well as irrigation [14]. The paper describes the types of social capital that exist in the field and the influence of this in mediating water insecurity. Section 5 concludes with the theoretical and policy relevant contributions of this research.

## 2. Conceptualising Social Capital and Power in a Peri-urban Context

Social capital is widely used in social sciences literature to refer to the quality of networks, social relations and ties that individuals mobilize in the pursuit of their objectives. It includes such ideas as the extent of civic engagement, norms of reciprocity, civic identity and social cohesion [10,11]. Within the context of natural resource governance, social capital captures the idea that social bonds and norms are an important aspect of natural resource management [11]. Based on this notion, social capital can be described as the relationship among actors [15]. It reduces the costs of collective action and facilitates and incentivises cooperation, building confidence to participate in activities, considering that others will do so as well.

In the natural resource governance literature, social capital is seen as being related to differential access to resources, and is understood to be a key concept explaining why different degrees of benefits can be yielded by actors with the same access to various forms of economic, political and cultural capital [7,8]. Governance literature has highlighted the benefits of social capital in terms of flow of information, gaining influence and control and building solidarity among a social collective [16–18].

Four ideas are critical to the concept of social capital. First, is the relation of trust between actors that reduces transaction costs and improves the efficiency of resource use. Second, is the element of reciprocity and exchanges, which increase trust. This aspect contributes to long-term obligations between actors and facilitates the achievement of collective positive outcomes. Third, the use of common rules and norms support the placing of group interests above certain powerful actors. Lastly, connectedness amidst groups may foster trading of goods, mutual help, and exchange of information between groups. In this context, [18] discusses the value of two-way connectedness. The concept of social capital is explained as a consideration within a social group that can be mobilized for collective action (or cooperation). This consideration can be in the form of sympathy, trust and forgiveness obtainable by various social relations [17]. The presence of social capital facilitates the process of social organization through networks, norms, and social trust, eventually facilitating coordination and

cooperation for mutual benefit. Lack of social capital may result in weak community development and collective action, making such communities powerless.

Social scientists have been explaining social capital through various definitions, though they tend to converge. However, one important difference among these definitions is the focus on external and internal relations of actors. External relations refer to certain actors interacting with the outside actors and internal relations refer to actors' interaction within a 'collectivity' (community, village, organization or a country) [16,18,19]. In this article, we mostly focus on internal relations, exemplifying how different members of a peri-urban community use social capital to improve water security. In particular, we focus on two aspects of social capital, trust and reciprocity, and exchanges between groups.

We analyse the social relations of power and the effect such relations have on controlling water resources [13]. We specifically focus on the role that power plays in accessing water and shaping water security. Taking from this perspective, it is important to understand what power is. We use [20] conception of power. Giddens (1984) defines 'power' as a 'transformative capacity' of an actor; any and every action that has the capacity to make a difference in the world can be considered as power. He further emphasizes that actors are not always aware of their influence and action that may go against many other actors' vested interests [16]. However, actor's power is usually translated or related to the held 'resources'. Giddens suggested two types of resources: allocative resources—that refer to controlling and owning physical or material resources to influence other actors, e.g., a factory, and authoritative resources—control over the activities of people, like civil services, or politics. For the purposes of this article, we use the concept of power that is translated or related to the authoritative resources, especially the ability to exercise opinion, control and influence by actors.

A community may have robust institutions based on social relations, but power relations between actors are dynamic and this may influence social relations [20]. For instance, caste hierarchy does play a role in shaping power relations [21]. Literature from South Asia has deliberated how caste hierarchy influences implementation of community forestry and irrigation systems [21–23]. However, in this article, we demonstrate how access to water is not directly related with caste hierarchy. We go beyond the aspect of leveraging social power to gain water access and highlight that positive influences of power and social capital can benefit communities by improving water security in peri-urban areas [24–26]. We relate the concept of social capital to 'cooperation' and empirically show how communities use cooperation to improve water security in peri-urban Gurgaon.

## 3. Material and Methods

Gurgaon is a major residential, outsourcing and recreation hub of the North-West Indian state of Haryana, that was transformed from a sleepy village of the 1980s through rapid real estate growth. There has been major land use change wherein land has been acquired by the government and private players for building gated residential areas, industrial parks, offices, malls and recreation centres. This was facilitated by Gurgaon's strategic location near the capital city of India, New Delhi and the international airport, barely 12 km away. The state enforced policies promoting private enterprise, which resulted in a visual landscape wherein high-rise gated communities and recreation malls exist alongside village residential areas.

Budhera village is located on the main highway (15A) that connects Gurgaon to the adjoining town of Farukhnagar, with a distance of 15 km from Gurgaon city. It has 900 households with a population of 5800 people. Harijans (a social group at the bottom of the caste hierarchy that are traditionally engaged in menial occupations such as scavenging) are numerically the most dominant social group, comprising 50 percent of the village households. The rest are divided amongst the Pundits, Yadavs (a high caste social group that own land and practice agriculture),and a few households of Balmiks (a social group low in the social hierarchy, some of them own land but a large majority rear livestock). Pundits are a landowning majority along with the Yadavs. The village has three Punjabi households too, settled after the India–Pakistan partition in 1947.There is a small group of Muslim migrants who live in the village, who are engaged as waste collectors for a local businessman.

The village has an interesting 'peri-urbanscape', being characterized by a mixed land use, increasing occupational diversification in the face of land acquisition and close links with the towns of Farukhnagar and Gurgaon, realized through a diversity of means of transport. The agricultural area of the village is about 2500 acres. Land is cultivated under a wide diversity of land tenure arrangements. Wheat and paddy are important rabi (winter) and kharif (monsoon) crops respectively, while fruit orchards owned by the urban elite are also common. Budhera could be termed a "land acquisition village" [5]. It has given land to the State for the construction of the Gurgaon Water Supply (GWS) Channel, the National Capital Region (NCR) channel, and the Budhera-Chandu water treatment plant, which shall be a primary water treatment center for the NCR.

Budhera was selected primarily because the residents of the village had lost land and water resources to build a water treatment plant to meet the needs of the city of Gurgaon. The intention was to study how the residents of Budhera had adapted to these changes. Gurgaon was an apt location for this study; it is projected as a millennium city, and as mentioned above, has undergone land use change to support urban expansion. Peri-urban spaces have lost land and water resources to support this expansion.

This study follows an ethnographic approach, using qualitative methods of data collection and analysis. Pursuing a case study method [27], we work on the premise of analytic generalization against statistical generalization. The goal is to generalize at the conceptual or theoretical level, against the goals of statistical generalization. Tools such as semi-structured interviews, group discussions, and direct observation were employed to collect data. For the purpose of this study, different communities were interviewed separately, so that their individual voices could be captured. A total of 50 interviews were conducted over a duration of two months (between December 2012–February 2013), with 8–10 interviews from each of the five castes.

We used snowball sampling as a technique for finding respondents for semi-structured interviews. Snowball sampling is created by a series of referrals made within a circle of people who know each other [28]. Women were interviewed separately in order to capture their experiences and perceptions. The analysis of the gendered dimensions of changing access to natural resources is presented in following sections of this paper; this is exemplified in the subordination of lower caste women by higher caste women in the tasks of water collection and the changing gender relations around livestock rearing and fodder collection consequent upon the acquisition of the grazing lands of the village. All interview texts were transcribed, coded and classified using key concepts such as social capital, caste hierarchies and allocative and authoritative resources. These were then subsequently translated into periodic memos that became the basis for analysis and theorization. To capture community's lived experiences, especially those of women and lower caste groups,we have used quotations, seeking to make the evidences and narratives powerful.

## 4. Historical Suffering and the Birth of an Alternative Power Structure

### 4.1. The Hierarchical Caste Structure and Its Effect on Water Availability

Like many other villages in the surrounding area, this village has developed a cluster system of living; particular communities live in particular places; the houses of Harijans are found in one area while that of Pundits in another (Figure 1). Though historically Budhera had a rigid caste structure, which exists till date, the caste system no longer has the same role in shaping social relations as it did earlier.

Drinking water is provided by the PHED, the Public Health Engineering Department, after being stored in two tanks prior to distribution in the village. A wastewater canal cuts through the village, carrying the domestic waste of Gurgaon city. The Government releases treated wastewater into the canal; farmers with adjoining fields are allowed to use it for irrigation. The farmers use this water for the growing of crops, especially paddy and wheat, as it is nutrient rich and reduces the need for the application of chemical fertilizers. Farmers access this water through plastic PVC pipes and diesel

engines. Two parallel fresh water canals cut through the village, namely, the NCR Channel and the GWS (Gurgaon Water Supply Channel). These canals carry fresh water for water treatment plants to meet the needs of the city of Gurgaon. The residents of Budhera are, however, not allowed to use the water from these canals. Nevertheless, hand-pumps are often installed along the banks of these canals to take advantage of the water table.

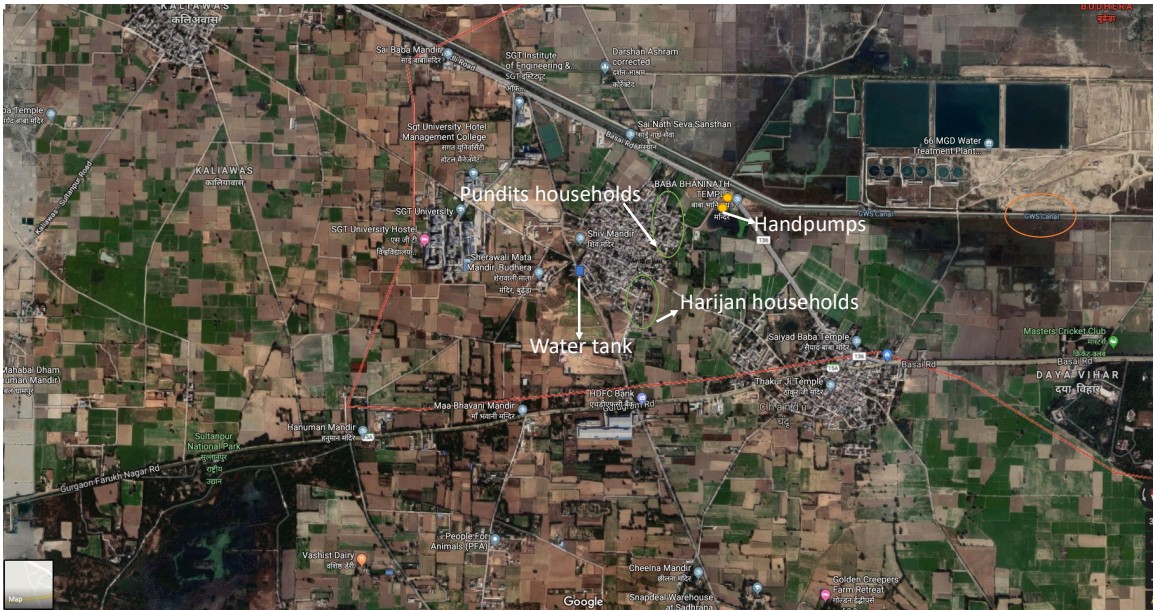

**Figure 1.** Map of village and key water sources; Source: Google maps.

Groundwater is still an important source of irrigation for some, though the groundwater is saline and borewell drilling is formally illegal (a High Court order in 2012 prohibited extraction of groundwater by setting up borewells for construction or residential purposes, allowing only civic agencies to withdraw water). Some trading of groundwater is prevalent. Certain relations built in the past have guided these phenomena. In earlier times, farmers would share tube-wells and the price of this groundwater sold would depend on social relations, like kinship ties, location in the village, or other such criteria. Some farmers are still dependent on it and their social ties have enhanced this system, but this dependence is also a measure of vulnerability. Lands are steadily being acquired in peri-urban spaces; as many farmers are dependent for their water on a single piece of land, the selling of such an important source of water may serve as a setback for many.

There is considerable variation in the water table level in the village, as well as in the quality of water. The water table level near the two fresh water canals is higher than that in other parts of the village.

In the absence of good quality water supply, the people of Budhera have to make do with two hand-pumps available to them. These hand-pumps are located near the GWS and NCR channels described above. In summers these hand-pumps become places of conflict and tension, as people have to wait long hours to fill their share; patience is at best fickle. "Dimag ghoom jata hai garmi mein" as a woman puts it (tempers flare in such heat).

The most vulnerable to this situation are migrant Muslims who migrated into Budhera three years ago. They came from Assam and are engaged as waste collectors for a local business man. They deal in plastic scrap. They live in makeshift tents; there is no official piped water supply for them. Their lack of secure land tenure translates into water insecurity.

Their access to water depends on the mercy of the Malik (their boss). This is characterized by much arbitrariness. Many times, due to lack of electricity the bore does not work. On days on which only a limited amount of water is pumped, the Malik's family claims first right to access the water.

Only after the needs of the Malik's family are met, do the migrants get access to water. This community adapts by storing water in plastic containers, whenever they get the opportunity to fill water, as they do not know when their turn will come next. Sometimes they have not been able to access water for three days, then they have to depend on the Balmiks who live close by and are not very stringent about religious or caste norms. Their vulnerability is heightened because they do not use the shared hand-pumps as they feel it will lead to problems in the village. According to them, they have not tried using them either, as they had been warned by the Malik to not hurt local sentiments. Since they are outsiders, they are prohibited from using the hand-pumps.

Traditionally, however, drinking water was mainly accessed from wells. Social status and water access were closely interwoven. A collective well belonged to each caste and one was not allowed to appropriate water from a well not belonging to one's caste. Further, areas of upper caste domination had more wells placed in them. There were five major wells, namely, (i) Bodiya; (ii) Pyau; (iii) Dungawala; (iv) Lal Kuan; and (v) HarijanKuan. These wells were named after the ancestors of the various social groups using them. They were located in different clusters; the lower castes were only allowed to use the Harijan Kuan. Lal Kuan was meant exclusively for the Pundits, and the remaining wells were used by all other castes, such as Yadavs. Though Harijans were numerically the most dominant caste they could only appropriate water from one well, whereas upper castes, though they were in a minority, were allowed access to water from four wells.

Several institutions, local norms, structured the relationship of people with water. These transformed over time, however, with the dilution of the caste system. For instance, wells and temples were maintained through the collection of "Dharamada" (Dharma means religion. Dharma is also a term that is used to describe an act of doing good. Dharmada refers to a contribution of charity with an intention of doing good). The patriarch of the family, in whose house either a son was born, or a marriage was going to take place, donated money; this constituted the Dharmada contribution. The Dharmada was collected by the Panchayat. The amount at the time of carrying out this research was INR 51 per contribution; village elders however recall that during their childhood days this used to be Rupee 1. After the Dharmada is collected, the patriarch is supposed to consult with the Panchayat on the most pressing issue whose resolution can be supported through the contribution of the Dharmada. Traditionally, Dalit families did not participate in this practice; however, they started doing so about 15 years ago. This points to the diluting effect of the caste system on the social structure and provides evidence of increasing social mobility.

In the early 90s, drinking water wells started to fall into disuse on account of a lowering of the water table; shared hand-pumps became the main source of drinking water. Though these hand-pumps were installed by an individual to earn religious merit, the whole community could access this source of water. Providing a source of water is seen as a noble deed: these hand-pumps commemorated the birth of a son or a marriage within the household. By doing so, the contributor earned "punya (merit)". These hand-pumps are installed by individuals as altruistic acts; however, their maintenance was supported through the collection of Dharmada collectively by the community. Most families could not afford to install new structures; they instead provided contributions towards maintaining the hand-pumps.

Water access and management have been socially embedded processes, with a close relationship between access to and management of water sources on the one hand, and social practices and institutions on the other. Lower castes would face discrimination, often having to wait in line for hours before their turn came. Often when they did fill water before the turn of an upper caste member, the latter would clean the hand-pump before filling his container with water, creating a long waiting period for those behind him in the que. This was because it was believed that the hand-pump had become impure by being touched by a lower caste member. Cleaning the hand-pump was an act of purifying the same, ridding it of its impurities.

In this system for taking water, moreover, lower caste women faced subordination by higher caste women. They had to take their pot and the upper-caste women would fill it with water for them.

The upper caste women arbitrarily decide the quantity of water to be taken back home by the lower caste women.

Some incidences of such phenomena are still seen in Budhera. However, making the lower caste wait longer in order to take water is not that common. While the new generations tend to mingle easily across castes, this is less common among elderly, more conservative individuals. As put forward by Tau Magan Ram, a prominent pundit from Budhera," woh zamana alag tha, bachpan se sikhaya gaya tha ki dalit se door raho, saath khana aurmilna uchit nahi hai, par ab baat kuch aur hai, ab chalta hai, par ab hum boodhe log abhi bhi puritarah se nahi milte julte, itni saalo ki seekh, ek minute mein toh nahi mitti"(Those times were different, we were as kids taught to avoid dalits, and were told that it was not appropriate to mingle and eat in their company, but now things have changed, the new generation openly mingles, but old people like me still are not very comfortable with proximity to dalits, we have believed this for far too many years to change our opinion now).

### 4.2. Land and Its Effect on Water

In 1962 Budhera went through what is known as Chakbandi, which refers to a process of land consolidation by the state. The Patwaris (local land record keepers) held absolute power over land records at that time. During this process, these Patwaris collaborated with the upper caste elite in manipulating land rates in exchange for bribes. This caused the land holding of many Harijan families to reduce in terms of physical size. For instance, Land A was for INR3 per acre, as it was underlain by sweet water. Land B was for INR1 per acre as it did not have any water source. Several Harijans claim that their A class land was exchanged with B quality land, though the new lands were marked under category A; thus, they were compensated with only 1/3rd of their previous landholding. In this process, arbitrary decisions were made without the requisite evaluation process being carried out: the same quality of lands were marked under several different categories; this created inequities after the Chakbandi was carried out. The land holding of the upper caste increased manifold. This was an important factor that gave a boost to livestock rearing in the village as it meant that many households were manipulated into becoming landless.

After being manipulated out of their fair share of land, the lower castes coped by switching to livestock rearing. They have, however, lost that source of income too. In 2008, the Government of India started to develop a 360-acre Budhera Chandu water treatment plant to meet the needs of the growing city of Gurgaon. This was done by taking away most of the grazing land that was used by the livestock rearers of Budhera. The youth of the village is seeking jobs in the cities, fully aware that their traditional occupation patterns have been decimated on account of land acquisition by the state. The elderly and middle aged, however, who are reluctant to relocate or to diversify occupationally, are quite vulnerable to these changes. The effect of losing their land resources is seen most significantly on the plight of women, whose already burdened schedule due to water collection duties and other tasks at home and on the fields faces further stress through additional tasks for fodder collection (see also [5]). They are now required to collect hara chara (fodder grass) for livestock, whereas earlier the livestock would be led to the grazing lands by men who did this task by means of hereditary occupation.

### 4.3. A New Regime

In this section we describe how after the elections, with a change of regime, power relations within different caste groups were altered and how this translated into a change in the access to water of the lower caste groups that earlier faced discrimination, as described in the preceding section. In, 2009, a Harijan Sarpanch come into power in Budhera. The lower castes responded to this change by overcoming decades of discrimination at the hands of the upper castes. They now directed their attention to improving their living conditions. Durga ji, a Harijan lady, responded to this situation thus "humara samaya aya, phir paani aaya." (Now it's our time, now we get water).

Today, Budhera is supplied water through three main tube-wells; only two of these are however functional. The tube-wells are operated by the Public Health Engineering Department (PHED) of

Gurgaon. They transfer the water into two overhead storage tanks, referred to in the local language as "Double Tunky". The quality of water supplied in Budhera is in general deteriorating, mainly because of the failure of the tube-wells. Out of the three tube wells, only one has water that is not salty.

The water after reaching the storage tanks is supplied throughout the village, using a network of pipes and valves. These pipes are installed with the help of the PHED. The local government officials claim that the quality of water being supplied passes all the regulations, but this does not get translated to clean water in the field. Thus, although the overall picture of good quality water might be deteriorating in Budhera, we do notice that certain pockets in the village are accessing good quality water in acceptable quantities. We explain this process below.

With the quantity of water being a constraint in these areas, a junior engineer of the PHED in consultation with the Sarpanch, assigns duties to an individual to operate the valves. These valves help in diverting the water towards different clusters at different points in time. Budhera has three main pipelines; the first caters to the Yadavs, Harijans, and a few lohars (a lower caste group that works with iron; they are blacksmiths). The second pipeline is meant exclusively for the Harijans, who are now politically powerful. The third is meant for the Om Nagar Dhani (a dhani is a settlement area away from the main settlement area of the village). As per a key informant, water from the two tube-wells is not mixed; rather it is transferred separately to the two different tanks and distributed as per will. Thus, the Yadavs by virtue of staying in the same line as the Harijan (Sarpanch) are entitled to good quality water. The pundits are served by the Om-Nagar Dhani line; the water transferred here is insufficient and of extremely poor quality. The line which serves Harijan areas exclusively also does not have a constant supply of non-saline water, depending on the availability; either of the tanks is used to give water to this area. Although a Harijan Sarpanch is in political power in Budhera, all Harijans are not treated equally, but compared to the water supplied to the Pundits (Om Nagar Dhani line), they are comparatively blessed.

The Harijans (lower caste) display power through the use of political position (authoritative resources) in the village. Being a dominant population in the village and with the government's decision to block the Panchayat seat for the reserved category has given them control and influence over village administration. In terms of quantity, there is water scarcity and therefore, a junior engineer of the PHED in consultation with the Sarpanch gives duties to a village man to be responsible for operating the valves; the person responsible for releasing water does so in a manner which is agreeable to the Sarpanch.

*4.4. "Bhaichara"—Social Capital*

While the Harijans use their authoritative resources to improve their water security, as described in Section 4.3, the Pundits do so by mobilising their social capital. In this section we describe the different ways in which social capital is mobilized to improve water security.

Social capital is productive [29]; it makes it possible to achieve certain ends that in its absence would not be possible to achieve; it is the positive leverage of close ties [21]. Building from this, we see that social capital shapes access to water in Budhera in three ways. Firstly, it works as an institution of pooling in financial resources for collectively accessing groundwater as described below in the case of the Pundits; secondly it works as a system of sharing based on criteria like kinship, location of houses, family agreements and promises; thirdly, social capital can be considered to be embedded in an institution of trust. We elaborate each of these aspects of social capital and their role in improving water security below.

The first type of social capital is seen in the way the Pundits have pooled in capital for the installation of a shared submersible pump. The community of the Pundits living in the centre of the village finds it extremely difficult to fill water from the hand-pump along the canal, mentioned in Section 4.3, as it is quite far. The Pundits adapted to this situation by pooling financial resources so as to collectively install a submersible on one family's plot of land. Pundits in Budhera are one large extended family. They drink water from personal submersibles and social capital plays an

important role amongst them. Water sharing amongst their clan is historically rooted. Many members of the Pundit community paid for a personal boring and submersible handset, in a particular Pundit household, on the premise that they shall all share the water. This is how Pundits in Budhera have adapted to water insecurity in terms of drinking water supply. They mobilize their network and social relations to pool financial resources to collectively adapt to water insecurity. This enables them to overcome their poor access to water by the state, and the constraints posed by the hand-pump being away from their settlement.

The second can be seen in the way boring wells, which as mentioned in earlier sections of this paper are illegal, are unearthed in the village, and how most people rather than just looking away when a boring is done, which is illegal in the eyes of state law, actively hide the activity. This constitutes an act of reciprocity: when one family carries out a boring, the other family keeps silent, on the premise that this will be reciprocated when the latter carries out a boring. As the government authorities fail to provide adequate water to the whole of the village, boring becomes a socially acceptable norm.

There is also a third way in which social capital plays a role in the mediation of water insecurity. In Haryana, as mentioned in earlier sections of this paper, the extraction of groundwater is banned by the High court. This has led to a new breed of businessmen, who take a charge, which includes the cost of the whole setup including the motor and the bribe they would have to pay to various government bodies, namely the police and the groundwater department. Long-term tenants are not offered such services, it is imperative that the boring entrepreneur knows of the family wanting this done from a long time, as one complaint to the wrong person can fully destroy the system. Officers at the groundwater department also do not see this breach as something very serious, it is felt that not allowing a person to access water is not acceptable; many times, the officials turn a blind eye, it is as if they at some level agree with the community. Thus, this practice acquires social legitimacy. A farmer mentioned, "paani nahi lene denge toh kheti kasie hogi, kheti ke liye paani lena mana nahi hona chahiye, aur peene ke liye toh bilkul hi nahi. Humari sarkar factory ko paani barbad karne se nahi rok paati, tabhi asie kaam hote hai" (groundwater extraction should be allowed for agriculture and if not for agriculture, then it should be allowed for drinking purposes at least, the government is not able to check water wastage in industries, and thus we all have to suffer).

## 5. Discussion & Conclusions

In earlier sections of this article, we described the growing attention to issues of water insecurity in peri-urban spaces. The peri-urban space is positioned to be on the receiving end of urbanization, characterized by a re-appropriation of land and water resources from the peri-urban to urban spaces. This article demonstrates that while studying issues of peri-urban water insecurity, it is important to pay attention to internal differences within communities as shaped by social and power relations. Further, different social groups mobilize social capital differently and exercise power in the mediation of water insecurity.

In this article, we respond to a key question of how social capital is mobilized to improve access to water. We use the concepts of power and social capital to explain how the actors interact in peri-urban Gurgaon, paying attention to which social groups are powerful and how the powerless use social capital to adapt to the changing resource access and usage. We found that it is imperative to understand caste as a strong institution and remedial measures should take into account the aspect of power relations based on caste, kinship and gender [30–32]. This means that efforts at improving access to water need to be based on an understanding of social differences within communities, rather than work from a stand point of blue-print, one-size-fits-all approaches.

Social capital serves as an organizing principle for members of the Pundit community in this village, and enables them to adapt to growing water insecurity by not only pooling financial resources, but also by evolving norms for sharing water. Social capital is seen in the way the Pundits have pooled capital for the installation of a shared submersible pump. They mobilize their network and social relations to pool financial resources to collectively adapt to water insecurity. This enables them to

overcome their poor access to wateras created by the state, and the constraints posed by the hand-pump being away from their settlement.

We found that the flow of water is determined by power relations in the village; those having a say in the Panchayat, namely the Harijans, are able to have their interests protected. This observation is supported by the analytical construct of the hydro-social cycle. In this view, water flow is seen to be a part of a social and political process rather than purely an aspect of natural flow [25]. Erik Swyngedouw states that "true scarcity does not reside in the physical absence of water in most cases, but in the lack of monetary resources and political and economic clout" [33]. Further, it substantiates that social differences and social capital are interlinked in the context of water security [34]. For example, the analysis suggests that social capital faces gender-specific limitations and more research and in-depth analysis is required to understand how benefits and losses of social capital are organized by gender in a peri-urban context [35–37].

In discourses on peri-urban water (in)security, peri-urban communities are seen as posited against the urban; this assumes peri-urban communities themselves to be a homogeneous group, losing control over and access to land and water resources. Analyses of peri-urban water insecurity need to be grounded in a study of social relations of power within peri-urban communities. Peri-urban scholars studying water insecurity should pay explicit attention to differential access to social capital in peri-urban spaces [38]. In the face of diminishing access to and control over natural resources, it is necessary to pay heed to how peri-urban communities mobilise social capital to maintain or improve their access to water, and how different social groups differ in their ability to do so. Interventions aiming at building 'community resilience' in peri-urban spaces need to unpack the notion of 'community' to understand social and power differences within communities that shape their differential access to water and the differential vulnerabilities to water insecurity.

**Author Contributions:** Conceptualization—V.N., S.V., A.D.; Methodology—V.N., S.V., A.D.; Validation—V.N., S.V.; Formal Analysis—V.N., S.V., A.D.; Investigation—V.N., A.D.; Writing—V.N., S.V., A.D.; Writing-Review and Editing—S.V., V.N.

**Funding:** This research was funded by International Development Research Center (IDRC), Canada, grant number [106248-001].

**Acknowledgments:** We would like to express our gratitude to SaciWATERs, South Asian Consortium for Inter-Disciplinary Water Resources Studies, Secunderabad, India, for organizational support during the study.

**Conflicts of Interest:** The authors declare no conflict of interest. The funders had no role in the design of the study; in the collection, analyses, or interpretation of data; in the writing of the manuscript, and in the decision to publish the results.

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
