# Peer review of "Bonds, Battles and Social Capital: Power and the Mediation of Water Insecurity in Peri-Urban Gurgaon, India"

_water, doi:10.3390/w11081607_

Round 1

Reviewer 1 Report

Overall an enjoyable read. 

My comments attached are mainly to clarify and sharpen the arguments.

Author Response

Map is inserted

It has been specified that the dharmada is collected by the sarpanch. 

The drinking water wells fell into disuse, not because of pollution, but because they ran dry. This has been specified in the paper in section 4, where we mention that they went dry because of the lowering of the water table. Indeed, they were shallower than the hand-pumps.

We do not have data on what fraction of the households have private borewells, as this would require a different research design (for instance, one based on a household survey, rather than the ethnographic approach adopted currently in the paper).  A discussion of the overall water situation of the village is however provided at the beginning of section 4 (section 4.1, para 2). The different sources of water are mentioned: the freshwater canals, the wastewater canal, hand-pumps and borewells and the water supplied by the state through the PHED (Public Health Engineering Department). Farming is indeed still practised in the village and so is water vending. Details of water vending, though practised only on a small scale, are presented in early parts of section 4. Details of major crops grown appear in early parts of section 4, when there is a mention of the wastewater being used for growing paddy and wheat. We mention the prevalence of land tenure arrangements as well as the total area under cultivation, to satisfy the reviewer's query about whether agriculture is still prevalent in the village, and also that the agricultural produce is sold in the wholesale markets in Gurgaon and Farookhnagar.

Of course, there is piped water supplied in the village. A big portion of section 4 of the paper is devoted to a discussion of the (power) dynamics surrounding that. 

The observation about the youth of the village seeking jobs in the city has been substantiated and the reasons for this reiterated (section 4). This is because their land holdings have diminished on account of land acquisition. Thus, push factors are at work.

The influence of the role of the new Harijan in correcting the imbalance in the allocation of piped water has been made further clearer in section 4. This is reinforced towards the end of section 5 to bring greater clarity.

It has been clearly mentioned in the paper that the migrants get water at the mercy of their boss. This is because it is only after the boss' own requirements are met, that they are allowed to get water. The bore is not pumped every day because of the erratic power supply. Only on days that there is power supply, is the bore pumped. Then, on these days, they get water only after the boss' requirements are met. 

Footnote number nine clarifies the legal status of borewell drilling in Gurgaon. We mention that there is considerable variation in the water table level in the village as well as in the salinity and water quality.

Given the reviewer's observation that the discussion of wastewater irrigation in the social capital section seems out of place, it has been removed from the paper. 

Though we have not collected caste-wise data on wealth and income, again, as that would require a different research design (e.g. survey), intuitively it could be asserted indeed that the pundits are on an average, wealthier. Historically, the Pundits had both types of Gidden's resources, now the Pundits have predominantly the second type, while the pundits the first, the migrant Muslims neither and the Yadavs are mid-way between both.   However, as mentioned in the early sections of this paper, we focus on authoritative power.

Reviewer 2 Report

This paper investigates water security in peri-urban India, and I agree with the authors that this geography requires much more investigation. However, this article is too diffuse in its concepts and too descriptive in its analysis. Let me comment on each of these separately, before noting some minor errors.

First, the introduction needs to be tightened so that the paper can move swiftly into the heart of theorizations of social capital and power. As stated in the introduction, the contributions of the paper jibe with gaps in knowledge per the literature cited. All is well. However, in moving to an explanation of social capital and power, and how these concepts will be used to frame the analysis, social capital comes to mean just about anything—relationships, trust, cohesion—all these are named in the paper and the reader is left wondering, “What isn'tsocial capital?” As deployed later in the discussion, social capital appears to mean cooperation. The offered meanings of power derive from strong sources (e.g., Giddens) but it isn't clear how power and social capital are different, and why it matters for understandings of water security that both be brought to bear on the case study. Critically, water in/security is never defined.

Second, although the authors build a dense scaffolding (albeit rickety) upon which to hang their evidence, they describe the situation on the ground in a rather straightforward, uncritical way. It is purely descriptive—too descriptive—and should be cut down to include only what is to be analysed through a return to the social capital analytic. Instead the authors take what they observed on the ground and fit it into structures, like caste, that are reported to have changed andnot changed in the recent past. Dalits taking control of village leadership appears as the most significant change influencing water security, as well as state-level decisions about water distribution. Why is it important to use social capital and power as a lens then, if driving forces, as described, are structural? (Why not use Giddens' structure and agency?) The evidence and the theory are not synthesized.

In short, the theory is opaque, the evidence and analysis descriptive, leaving the explanation of current conditions of water security rather dull. Sharpening the analysis to a cutting edge requires refinement of the concept of social capital (or power, or structure/agency, or access, or practices, or infrastructure—but only one of these) and of water security. Only then can the data analysis contribute to new ways of thinking about water security and social relations. How is caste produced and reproduced, and what does water have to do with that? Why cannot citizens pressure the PHED to act? How will we know when this village and these communities are water secure? Is that even possible under current political (at multiple scales) conditions?

Minor points:

The methods section is solid but requires information as to how/why Gurgaon, the village, and the participants were selected. This is especially important if the authors want to analyze caste relations.

The addition of other types of water besides drinking water at the end of the paper is confusing.

The definitions of the caste groups are misleading, e.g., Dalit v. Harijan; Brahmins described as 'upper' when in fact they are 'uppermost'.

Women were interviewed but gender is not analysed. The authors’ use of the term ‘lady’ is anachronistic.

Some problems with English usage, e.g., line 33 Introduction: “less insights” should be “fewer insights” as ‘few’ is an adjective of quantity for countable nouns. 

Below are some sources that may prove useful to the authors in their revisions.

Das, P. (2014). Women’s Participation in Community-Level Water Governance in Urban India: The Gap Between Motivation and Ability. World Development64, 206–218. https://doi.org/10.1016/j.worlddev.2014.05.025

Ekers, M., & Loftus, A. (2008). The power of water: developing dialogues between Foucault and Gramsci. Environment and Planning D: Society and Space26(4), 698–718. https://doi.org/10.1068/d5907

Joshi, D. (2011). Caste, Gender and the Rhetoric of Reform in India’s Drinking Water Sector. Economic and Political Weekly, 56–63.

Joshi, D., & Fawcett, B. (2005). The role of water in an unequal social order in India. In A. Coles & T. Wallace (Eds.), Gender, water and development(pp. 39–56). New York, NY: Berg.

Mehta, L., & Karpouzoglou, T. (2015). Limits of policy and planning in peri-urban waterscapes : The case of. Habitat International48, 159–168. https://doi.org/10.1016/j.habitatint.2015.03.008

Motiram, S., & Osberg, L. (2010). Social capital and basic goods: The cautionary tale of drinking water in India. Economic Development and Cultural Change59(1), 63–94. Retrieved from http://www.scopus.com/inward/record.url?eid=2-s2.0-77957806243&partnerID=40&md5=2df5bcfe8089b195e10c3a378d432c51

Narain, V. (2016). Whose land ? Whose water ? Water rights , equity and justice in a peri-urban context,9839(February). https://doi.org/10.1080/13549839.2014.907248

O’Reilly, K., & Dhanju, R. (2014). Public taps and private connections: The production of caste distinction and common sense in a Rajasthan drinking water supply project. Transactions of the Institute of British Geographers39(3), 373–386. https://doi.org/10.1111/tran.12039

Randhawa, P., & Marshall, F. (2014). Policy transformations and translations : lessons for sustainable water management in peri-urban Delhi , 32, 93–107. https://doi.org/10.1068/c10204

Truelove, Y. (2011). (Re-)Conceptualizing water inequality in Delhi, India through a feminist political ecology framework. Geoforum42(2), 143–152. https://doi.org/10.1016/j.geoforum.2011.01.004

Wallace, T., & Coles, A. (2005). Gender, water and development. Berg.

Author Response

We thank the reviewer for his endorsement that the issue of water security the periurban geography of India requires much more investigation. We have attended to the concerns raised by the reviewer regarding the concepts of social capital, power and water security.  The concepts of power and social capital and the relationship between the two is elaborated more carefully in the introduction (section 1). These concepts are more clearly elaborated in section 2 and their operationalization has been further explained more minutely. We emphasize that we focus on two aspects of the components of social capital, namely, the element of cooperation and the norms of reciprocity. While pointing out that social capital refers to both internal and external relations, we emphasize our focus on internal relations. 

The evidence and theory are now better synthesized through the discussions in sections 1 and 2 and then reinforced again in section 4 and in the conclusions in section 5. 

In the methods section, we have provided a justification for the selection of Gurgaon for this study. Given the reviewer's observation that the addition of the description of other sources of water than the piped water supply is confusing, we have brought that discussion earlier in the paper, in section 4, para 2. 

Gender is analyzed at several points in the paper, for instance, while referring to the implications of the acquisition of grazing lands for the division of labour around livestock rearing (lines 322-327) and the subordination of lower caste women by higher caste women while in queue for taking water (lines 287-290).  This is also summarized in the material and methods section, given the reviewer's concern for greater clarity. Given the reviewer's observation that the use of the word 'lady' is anachronistic, we no longer use that word. "Less insights" is replaced by "fewer insights" in section 1, para 2. 

Reviewer 3 Report

The central theme and the argument made by the authors (social capital in peri-urban areas and its relationship with power and access to water can indeed make valuable contribution to the world of development praxis. Much has already been written about social capital in rural and urban setting. It is quite adequately covered in this paper too. However, I was looking forward to a discussion on whether it will require significant adaptation (from both epistemology and ontology)  to a peri-urban setting. Section 2 raised my hopes, but I am not sure if I got much out of it.That would have been a significant contribution. Given its transient temporal and spatial dimensions and the socio-political-economic complexities, a better understanding of social mobilization process and the impact on local governments in peri-urban areas would be a valuable. The discussion in how caste and class hierarchies shape the ability to foster collective action (and hence social capital) is useful. Some confusions can be avoided --- is this a case in point for strengthening the representation of various interest groups in the local government (details of which is important, but covered inadequately) or just focussed on dealing with legacy issues of caste and class divisions. The notion of one-way reciprocity (line 92) comes across as an oxymoron.

The paragraph 4 section 2 (line 108 to 119) was meant to be a discussion on social capital and power from "authoritative" resources. Some sentences could be explained better: for example line 112 is questioning whether social capital influences collective action? That relationship is firmly established, I would think! another example is line 115: Economic resourcefulness is key to adaptation strategy -- that is almost a tautological statement. Therefore the fact that Pundits are better in mobilization should not come as a surprise. The fact that they are from one extended family further helps. I would not conflate that with higher social capital among Pundits. 

The historical backdrop to Budheda village could be more useful if the chronology of events was tied to the year it happened. Does the narrative cover 25 years, 50 years or 100 years? The data on community wells is insightful, but information on privately owned wells and tube-wells/hand-pumps and government sponsored water schemes is sparse. I believe the analysis of ownership and access to various sources will be quite revealing. A clear picture of the extend and nature of discrimination and even brazen exploitation in the village does not emerge. Some descriptions indicate quite serious problems and other stories indicate that the divisions are now more benign. In the absence of that information, it is difficult to draw any conclusions about the rather dramatic transition to a Harijan controlled Panchayat. Has this change enabled better social mobilization process? Or does this new found "power" explain why the new water supply systems appear to swing the advantage towards the non-upper caste households. The entry of Muslims and the role of their "Maliks" appears to have some impact on access, but I could not infer much.

It will be a good idea to compile, perhaps in a table, all the water sources over the years and show a pattern in their ownership. We know that property rights have a profound influence on how assets are created, used and maintained.And when some of those assets are communally owned, the dynamics become even more complex. 

The conclusion and policy imperatives are not very convincingly argued on the basis of the case study. For example, I re-read the paper a few times to see if there is any inference drawnon whether social capital formation actually happened. If it did, was it adequate (which would require a discussion on what is adequacy). The three policy insights do not address the central theme. The first one talks more about the need for strengthening democratic ethos (in my opinon the adequacy of the modes of representations in the local government rather than the need to catalyse collective action across caste and class divisions. The second insight is meant for a resource manager rather than a social mobilizer -- it talks about the need to have an integrated approach to both supply and demand. The third one does point out the need to better understand the power relationships before social capital formation can happen. What I would have liked to see is a synthesis of what was found in the case and how does it help explain the central theme : social capital and access to resources in peri-urban areas.

Author Response

We appreciate the reviewer's observation that there is much scope for knowledge creation on the role of social capital in periurban spaces, though much is written already about social capital in rural and urban contexts. In the introduction, we have made a more specific case for arguing what is unique to the analysis of social capital in periurban contexts. In other words, what is it about the periurban that makes social capital a concept of interest? We provide an argument to the effect of why it is necessary to study social capital in periurban contexts in the revised version in section 1, lines 46-53. The notion of one-way reciprocity, (line 92, previous version) is indeed an oxymoron and has been removed in the revised version.

Again, a detailed analysis of the ownership patterns of different sources of water would require a different research design than what has been adopted in the paper. 

Following the observation that the three critical policy insights actually do not flow from the empirical basis of the paper, the conclusion section has been rewritten. The conclusions section is revisited to focus on the key learnings from the study about the role of social capital and power. Indeed, there is no evidence in the paper to substantiate the claim of whether social capital formation actually took place. The conclusions now reinforce the theme of the paper on the relationship between social capital and access to resources in periurban areas. The policy-relevant message is that power differences among communities need to be recognized in capacity building programmes in periurban areas that seek to build 'community resilience". The concept of 'community' needs to be unpacked to recognize internal power differences, rather than seeing periurban 'communities' as a homogeneous whole posited against 'urban' communities, to whom they lose land and water resources. 

Following the reviewer's observation that the discussion of the various water sources at the end of the paper is confusing, this description is now brought earlier in the paper. This also serves to provide a discussion of the context of the research. 

Author Response

Literature review

We thank the reviewer for reviewing the manuscript and emphasizing that the issue of water security the periurban geography of India requires much more investigation. We have attended to the concerns raised by the reviewer regarding the concepts of social capital, power and water security. The suggested literature is used appropriately in the manuscript and the text for sub-comments 2 and 3 are revised. 

Methods/design

It is mentioned that data collection took place over 2 months, between December 2012 and February 2013. 

Did you come up with subcategories within the umbrella of social capital? (e.g., Trust and reciprocity and exchanges between groups.). How did you differentiate them? 

Based on the conversations during the interviews, we presented the analysis for trust and reciprocity. For example, if Pundits believe the Harijans or not. 

As suggested the language of the manuscript has been revised. 

A map has been inserted to explain the context of this study and how different water sources are used. 

Results 

We attempted to explain the elements of social capital (trust and reciprocity) as best as possible. Some of the queries have been responded by revising the text in the conceptual framework and results sections. However, we felt that including more literature will make the conceptualization more complex. 

The text regarding the borewells has been revised. 

The discussion and conclusion section has been revised accordingly. 

Minor corrections suggested by the reviewer have been made for improving the consistency of the manuscript.  

Round 2

Reviewer 3 Report

many of the concerns flagged earlier appears to be fully or partially addressed. I am fine with the revisions.

Author Response

We thank the reviewer 3 for accepting the changes made by the authors. 

This manuscript is a resubmission of an earlier submission. The following is a list of the peer review reports and author responses from that submission.